# How well are marginalised groups represented in electronic records? A codelist development project and cross-sectional analysis of UK electronic health records

Tetyana Perchyk ,[1] Isabella Joy de Vere Hunt ,[2] Brian D Nicholson ,[2] Luke Mounce ,[3] Kate Sykes,[4] Georgios Lyratzopoulos ,[5] Agnieszka Lemanska ,[1,6] Katriina L Whitaker,[1] Robert S Kerrison[1]

For numbered affiliations see end of article.

**Correspondence to**
Tetyana Perchyk;
t.perchyk@surrey.ac.uk

## ABSTRACT

**Objectives** Primary care electronic health records provide a rich source of information for inequalities research. However, the reliability and validity of the research derived from these records depend on the completeness and resolution of the codelists (ie, collections of medical terms/codes) used to identify populations of interest. The aim of this project was to develop comprehensive codelists for identifying people from ethnic minority groups, people with learning disabilities (LDs), people with severe mental illness (SMI) and people who are transgender.

**Design** We followed a three-stage process to define and extract relevant codelists. First, groups of interest were defined a priori. Next, relevant clinical codes, relating to the groups, were identified by searching Clinical Practice Research Datalink (CPRD) publications, codelist repositories and the CPRD Code Browser. Relevant codelists were extracted and merged according to group, and duplicates were removed. Finally, the remaining codes were reviewed by two general practitioners (GPs).

**Setting** The curated codelists were compared using a representative sample in the UK. The frequencies of individuals identified using the curated codelists were assessed and compared with widely used alternative codelists.

**Participants** Comprehensiveness was assessed in a representative CPRD population of 10 966 759 people.

**Results** After removal of duplicates and GP review, codelists were finalised with 325 unique codes for ethnicity, 558 for LD, 499 for SMI and 38 for transgender. Compared with comparator codelists, an additional 48 017 (76.6%), 52 953 (68.9%) and 508 (36.9%) people with LD, SMI or transgender code were identified. The proportions identified for ethnicity, meanwhile, were consistent with expectations for the UK (eg, 6.50% Asian, 2.66% black and 1.44% mixed).

**Conclusions** The curated codelists are more sensitive than those widely used in practice and research. Discrepancies between national estimates and primary care records suggest potential record/retention issues. Resolving these requires further investigation and could lead to improved data quality for research.

## STRENGTHS AND LIMITATIONS OF THIS STUDY

⇒ This study followed standardised methodology for developing codelists, enhancing the scientific rigour, transparency and reproducibility of the study.

⇒ Codelists were validated against existing codelists, using a representative sample of primary care medical records, comprising >10 million patients.

⇒ Only two general practitioners reviewed the codelists, neither of whom was an expert in learning disabilities or severe mental illness. As such, it is possible some conditions were overlooked or under-represented.

⇒ Comparisons with existing codelists were not like-for-like due to differences in population definitions.

## INTRODUCTION

Health inequalities are unfair and avoidable differences in health that are systematically established and arise from the social conditions in which people are born, grow, live, work and age.[1]

In the UK, addressing health inequalities is a national priority.[2] The National Health Service for England (NHS England) developed 'the CORE20Plus5 Framework', which outlines priority groups and clinical areas requiring accelerated improvement.[3] The 'Core20' refers to the 20% most deprived areas in England, while the 'Plus' refers to population groups that should be identified at a local level, including (but not limited to) ethnic minority groups, people with a learning disability (LD) or autism, people with multiple long-term health conditions, other groups that share protected characteristics (as defined by the Equality Act 2010[4]) and groups experiencing social exclusion (such as coastal communities, where there may be

small areas of high deprivation hidden among relative affluence).[3] The '5' describes key clinical areas requiring accelerated improvement for these populations, namely maternity, severe mental illness (SMI), chronic respiratory disease, early cancer diagnosis and hypertension case-finding and management.[3]

While the CORE20Plus5 framework provides a useful outline of national priorities for health research and clinical practice, there are several ongoing challenges that need to be addressed for researchers to support the national strategy.[5] Pertinent among these is the identification and classification of CORE20Plus5 groups (also referred to as 'socially disadvantaged' or 'marginalised' groups) within electronic health records (EHRs).[5] EHRs include key administrative and clinical data relevant to a person's care, including demographics, progress notes, problems, medications, vital signs, past medical history, immunisations, laboratory data and radiology reports.[6] These data are recorded using clinical coding systems, typically containing hundreds of thousands of concepts,[7] which need to be categorised via codelists (ie, collections of medical terms/codes) to facilitate research.

While methods for codelist development and reporting have previously been published,[8–10] and many studies now report the methods used to develop codelists for various medical diagnoses, few papers describe the systematic development of codelists specifically for identifying marginalised populations.[9] Some papers have attempted to develop codelists for certain marginalised groups, including those for identifying the autistic population,[11] the lesbian, gay and bisexual (LGB) population[12] and people who experience multiple long-term conditions.[13] However, there do not appear to be any papers describing the development of codelists for other marginalised groups.

To support future health inequalities research, we set out to develop comprehensive codelists for the identification of four CORE20plus5 groups, namely people from ethnic minority groups, people with LDs, people with SMI and people who are transgender. We selected these groups because there is a general lack of standardised methods for codelist development to describe these inequality populations in EHRs.

## METHOD

Previously published approaches for codelist development were reviewed,[9 10] and Watson *et al*'s[14] approach was deemed the most appropriate because it specifically sets out methods for developing codelists using UK-based EHRs.

The approach comprises three steps: *Step 1*: clearly define the clinical feature of interest a priori; *Step 2*: assemble a list of codes that may be used to record the clinical feature and *Step 3*: expert review of codes.

The following presents a detailed overview of the processes undertaken for each step.

---

**Box 1    The UK census ethnic categories.[15]**

**Asian or Asian British:**
   Indian.
   Pakistani.
   Bangladeshi.
   Chinese.
   Any other Asian background.

**Black, black British, Caribbean or African:**
   Caribbean.
   African.
   Any other black, black British or Caribbean background.

**Mixed or multiple ethnic groups:**
   White and black Caribbean.
   White and black African.
   White and Asian.
   Any other mixed or multiple ethnic background.

**White:**
   English, Welsh, Scottish, Northern Irish or British.
   Irish.
   Gypsy or Irish Traveller.
   Roma.
   Any other white background.

**Other ethnic group:**
   Arab.
   Any other ethnic group.

---

### Step 1: clearly define the clinical feature of interest a priori

Watson *et al* recommend beginning by clearly defining the clinical feature of interest.[14] To do this, they recommend using reliable sources of information, such as the National Institute for Health and Care Excellence.[15] For the purposes of this study, we used the following definitions and sources to define marginalised groups of interest:

#### Ethnicity

The Office for National Statistics[16] has highlighted that there is no true consensus on what defines an ethnic group, as identification with these is 'self-defined and subjectively meaningful to the individual'. However, as stated by NHS England, it is generally accepted that ethnicity includes a variety of elements, such as ancestry, culture, identity, religion, language and physical appearance.[17]

We defined ethnicity according to the characteristics set out in the Equality Act 2010, namely colour, nationality and ethnic or national origins.[4] To this end, the UK Census 2021 categories were used to group ethnicity codes (see box 1).[18]

#### Learning disability

The Department of Health and Social Care (DHSC)[19] defines an LD as 'a significantly reduced ability to understand new or complex information, to learn new skills (impaired intelligence), with a reduced ability to cope independently (impaired social functioning), which started before adulthood.' According to Public Health

England,[20] 'a learning disability is different from a learning difficulty, which is a reduced intellectual ability for a specific form of learning and includes conditions such as dyslexia (reading), dyspraxia (affecting physical co-ordination) and attention deficit hyperactivity disorder.'

We defined LDs according to the characteristics described by the DHSC. This is because there is a significant overlap between a diagnosis of LD and learning difficulty, and these two terms have historically been used interchangeably. To capture the most accurate population experiencing inequalities and to avoid possible misclassifications, the more inclusive DHSC definition was used to describe the LD population. This definition for LD excludes autism, which is important, as while many people with LD have autism, not everyone with autism has LD. Accordingly, no autism codes were included in the LD codelist.

We did not categorise LDs into groups. They are often described as being 'mild', 'moderate' or 'severe', 'verbal' or 'non-verbal' and 'syndromic' or 'non-syndromic', but there is no consensus as to which LDs fall into which categories, and there is individual-level variation within the same disability, making it inappropriate to generalise.[21]

### Severe mental illness

The Diagnostic and Statistical Manual, 5th edition (DSM-5), defines SMI as 'a mental, behavioural or emotional disorder resulting in serious functional impairment, which substantially interferes with or limits one or more major life activities.'[22]

We defined SMI according to the characteristics described in the DSM-5 and categorised codes according to four recognised conditions, namely 'schizophrenia', 'bipolar disorder', 'severe major depression' and 'other psychotic disorders'.[22]

### Transgender

The UK Government Equalities Office[23] defines trans (transgender) people as 'people whose gender is different from the gender assigned to them at birth.' It provides the following by way of example: 'a trans man is someone who transitioned from woman to man'.[23]

We used the definition provided by the Government Equalities Office.[23] Most clinical codes do not differentiate between trans men and trans women; therefore, we did not group codes into these categories.

### Patient and public involvement

Five members of the public with lived experience of autism, SMI, minoritised ethnicity and/or being transgender contributed to the study. While no one with LD was included, a person with autism was able to provide perspective on the differences between autism and LD, and their lived experience of the two being conflated with one another.

Initial plans and findings were shared with these collaborators during quarterly meetings. During these meetings, collaborators provided feedback on the codelists, the language used to define them and how to handle outdated or offensive terminology (such terms were retained within the codelist to ensure completeness but were omitted from the manuscript, so as not to perpetuate their use in medical literature.

### Step 2: assemble a list of codes that may be used to record the clinical feature

In the second step, an online thesaurus was used to develop a list of synonyms associated with the clinical features of interest. An overview of the synonyms used for each clinical feature is presented in table 1.

These identified synonyms were used to search literature and databases to find relevant published codelists related to the inequality groups. One major resource used was the October 2023 Clinical Practice Research Datalink (CPRD) Bibliography.[24] CPRD is one of the largest research databases of electronic, anonymised and longitudinal medical records from primary care in the world.[25] The CPRD Bibliography is a resource that provides a record of all published studies (>3000 as of October 2023) using CPRD data, and this aided in our search for publications. The relevant publications were identified using our list of synonyms, and the codelists used in these publications were subsequently downloaded where they had been made available.

To capture codelists used in non-CPRD studies, we also searched and downloaded relevant codelists from four widely used codelist repositories, namely OpenCodelists,[26] the Health Data Research UK Phenotype Library,[27] the University of Cambridge Codelist Repository[28] and

| Table 1 | List of synonyms used for each clinical feature |
| --- | --- |
| **Clinical feature** | **Synonyms** |
| Ethnicity | Nationality, nationalities, nation(s), race(s), racial, colour, ethnic, ethnicity, minority and minorities |
| LD | Intellectual disability, intellectual disabilities, learning disability, learning disabilities, ID(s) and LD(s) |
| SMI | Serious mental illness, serious mental illnesses, severe mental illness, severe mental illnesses, SMI(s), Schiz*, psychosis, bipolar and major depress* |
| Transgender | Trans, transsexual, transgender, gender dysmorphia, trans man, trans woman, trans men, trans women, transfeminine, transmasculine, gender surgery and LGBT |

*indicates use of truncation for searching variants of the root term e.g. Schiz* covers variants including Schizophrenic, Schizophrenia, etc.
ID, intellectual disability; LD, learning disability; LGBT, lesbian, gay, bisexual and transgender; SMI, severe mental illness.

the London School of Hygiene and Tropical Medicine Data Compass.[29]

Once all relevant published codelists were identified, they were combined into one Excel file for each inequality group. This gave us four files: one for ethnicity, LD, SMI and transgender. Within each of these files, any duplicate codes were removed and the remaining codes were grouped by the two main coding systems used in the UK: Read and Systematised Nomenclature of Medicine Clinical Terminology (SNOMED CT). Read was generally used in primary care until 2018, and SNOMED CT, thereafter, following the enablement of SNOMED CT within primary care systems.[30] This provided an aggregated list encompassing every code (excluding duplicates) that has been used and published either through the CPRD literature or through an online codelist database.

These aggregated codelists were then checked for accuracy using the CPRD Code Browser, which is a tool specific to CPRD and is provided to researchers with access to its data. This tool contains the diagnostic description, the alphanumeric Read/SNOMED CT code originally used by the general practitioner (GP) to enter the clinical information and CPRD's proprietary 'medcode'/'medcodeid' (which is simply a numeric equivalent of the Read/SNOMED CT code).[30] The tool allowed for easy browsing by using either a known code or description to check code accuracy. Searching by the medical descriptors for the code, or 'terms', was useful because each search provided a list of all possible codes. For example, a search for 'psychosis' showed every diagnostic term and associated code available from primary care GP records. This was useful to cross-reference each of the individual codes and ensure all relevant options have been included. In line with the method described by Watson et al,[14] inaccurate codes were removed or corrected, and additional codes (not previously identified) were added if they were associated with the diagnostic definition of one or more eligible codes.

### Step 3: expert review of codes

In the final step, the codelists were sent to a GP (IJdVH) for review. The codes were scored using the following three-point scale:

1. Definitely include: the code accurately defines the clinical feature of interest, and GPs would use it.
2. Uncertain: it remains unclear whether the code accurately reflects the clinical feature of interest or whether GPs would use it.
3. Definitely exclude: the code does not define the clinical feature of interest, and GPs would avoid it.

Codes assigned a score of 2 ('Uncertain') or 3 ('Definitely exclude') were reviewed by a second GP (BDN). All codes assigned a score of 3 ('Definitely exclude') by both reviewers were removed from the codelists. All codes receiving a score of 2 from either reviewer were retained as 'uncertainty variables', which are recommended for sensitivity analyses.[14] These variables are typically less frequently used by GPs but may identify additional, potentially relevant, cases.[14]

### Assessing codelist comprehensiveness

To explore the comprehensiveness of the developed codelists, we used the CPRD database to ascertain the number of people with at least one recording of an included code for each list. We restricted the population to those who had been registered with their CPRD Aurum practice since at least 1 January 2019 and who had neither died nor deregistered from their practice by August 2022 (the date of last data collection).

In the context of CPRD studies, researchers have been able to acquire already categorised ethnicity data through linkage with the Hospital Episode Statistics (HES).[31] Derived ethnicity data are also now available through CPRD at an additional cost.[32] The ethnicity codelist we generated will be useful to those researchers who are using other non-CPRD datasets or those who may not have access to either the linked HES data or the CPRD-derived ethnicity.

For the other variables, validated codelists from other sources were selected as comparisons, seeking to match the scope of each list as closely as possible. Codelists for LD and SMI were sourced from the Quality and Outcomes Framework (QOF: the pay-for-performance scheme for general practices in the UK).[33] QOF includes incentives for practices to keep a register of patients with certain conditions, including LD and SMI, and produces validated codelists for this purpose. We used lists from the QOF Business Rules V.45, 'Learning Disability codes' (list name LD_COD) and 'Psychosis and schizophrenia and bipolar affective disease codes' (list MH_COD) as our comparison lists. For the transgender category, no QOF list exists, and the literature is slim. We identified two identical previously published codelists that were used to compare our newly developed codelist.

### RESULTS

A review of 87 relevant publications and additional codelist repositories identified a total of 55 codelists: 16 for ethnicity, 16 for LD, 21 for SMI and 2 for transgender. After removing duplicates and expert review, 388 codes were excluded, and 1420 total unique codes remained: 325 codes for ethnicity, 558 codes for LD, 499 codes for SMI and 38 codes for transgender. The full review process is demonstrated in figure 1.

The codes were run in the source population of CPRD (n=10 966 759) to ascertain the number of people captured by the codelists (see table 2). People were identified by the presence of one or more codes related to the inequality group of interest. Ethnicity data were available for 4 384 017 (40.0%) individuals. Of these, 3 112 319 (71.0%) were ethnically white; 285 126 (6.50%) were ethnically Asian; 116 570 (2.66%) were ethnically black; 63 024 (1.44%) were of mixed ethnicity; 94 693 (2.16%)

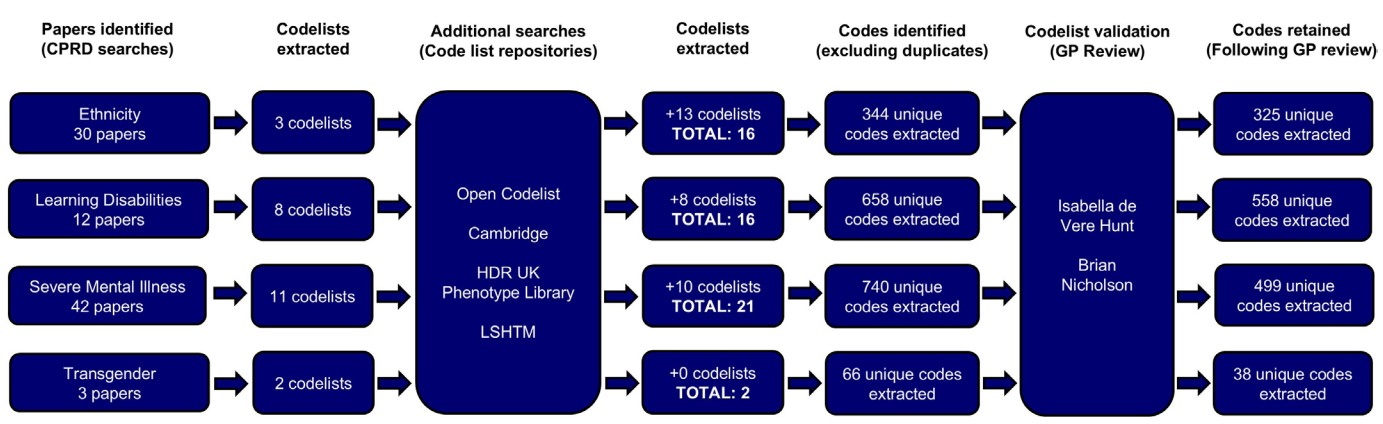

**Figure 1** Process outlining codelist review and development. CPRD, Clinical Practice Research Datalink; GP, general practitioner; HDR UK, Health Data Research UK; LSHTM, London School of Hygiene and Tropical Medicine.

were categorised as 'other;' and 712 286 (16.3%) were of unknown ethnicity.

110 692 people were found to have LD (1.00%), 129 788 had SMI (1.18%) and 1884 were recorded as transgender (0.02%). This represented an increased capture of 48 017 (76.6%) for LD, 52 953 (68.9%) for SMI and 508 (36.9%) for transgender (see table 3).

## DISCUSSION
### Summary

We created comprehensive codelists for identifying marginalised groups of interest, including ethnicity, LD, SMI and transgender records in EHRs. Compared with the commonly used alternatives, our codelists improved data capture by 76.6% for LD, 68.9% for SMI and 36.9% for transgender records. These codelists have been made publicly available for others to use in their research: https://osf.io/8skze/.[34] They are readily available for use in UK research but need to be adapted for international research, where different code systems are used. We provide code descriptions alongside the code to help facilitate this process.

### Strengths and limitations

This study has several strengths. First, codelists were identified through multiple sources, including CPRD publications and code list repositories, maximising the chances of identifying potentially relevant codes. Second, instead of accepting codes at face value, an expert review was implemented, with potentially irrelevant codes removed or segregated as uncertainty variables. The segregated codes will aid researchers for use in sensitivity analyses if they find it relevant to their projects.

This study also has a number of limitations. First, it only used codes from codelists arising from CPRD publications or uploaded to codelist repositories, and, despite several safety netting procedures, it is possible that not all relevant codes were captured. This is exemplified by the fact that only 29% of relevant publications made their codelists publicly available, suggesting that there is potential for some codes to be missed. Second, while we compared the comprehensiveness of our codelists against commonly used alternatives (eg, QOF), the comparisons were not like-for-like due to differences in population definitions. For example, the QOF definition of SMI does not

| Table 2 | Descriptives for CPRD population, including frequencies between ethnic groups derived from our curated codelist | | | |
|---|---|---|---|---|
| **Population total (n)** | **Mean age (SD)** | **Age range (years)** | **Frequency males (%)** | **Frequency females (%)** |
| 10 966 759 | 43.3 (22.8) | 4–100 | 5 554 024 (50.6) | 5 412 735 (49.4) |
| **Ethnicity** | **Frequency from curated codelist (n)** | | **Frequency percent (%)** | |
| White | 3 112 319 | | 71.0 | |
| Asian or Asian British | 285 126 | | 6.50 | |
| Black, black British, Caribbean or African | 116 570 | | 2.66 | |
| Mixed or multiple ethnic groups | 63 024 | | 1.44 | |
| Other ethnic group | 94 693 | | 2.16 | |
| Unknown | 712 286 | | 16.3 | |
| CPRD, Clinical Practice Research Datalink. | | | | |

**Table 3** Frequencies of populations extracted from CPRD, comparing original and curated codelists for LD, SMI and transgender groups

| Inequality group | Original codelist (n) | Curated codelist (n) | Difference in identification (n) | Per cent increase (%) |
|---|---|---|---|---|
| LD | 62 675 | 110 692 | 48 017 | 76.6 |
| SMI | 76 835 | 129 788 | 52 953 | 68.9 |
| Transgender | 1376 | 1884 | 508 | 36.9 |

CPRD, Clinical Practice Research Datalink; LD, learning disability; SMI, severe mental illness.

include severe major depression, and these differences may partially explain some of the increase observed in our codelist. Finally, only two experts reviewed the codelists, both of whom were GPs. While this was a strength in terms of them being representative of the population responsible for entering medical codes in primary care clinical systems, they may have been less familiar with the plethora of medical diagnoses associated with some conditions, such as SMI and LD, which may have resulted in some conditions being omitted from our codelists. We encourage experts (eg, LD nurses) to provide feedback and help refine these lists.

### Comparison with previously published literature

This research builds on the extant literature by synthesising the efforts of those who have previously curated codelists for the identification of marginalised groups. For example, Boyd et al[35] previously created a code list for identifying people who are transgender, which included a total of eight Read codes. Our list includes an additional 30 Read codes (along with the corresponding SNOMED CT codes), enabling higher resolution and identification of people who are transgender.

### Implications for research

The findings of our study have several implications for research. Since only a small proportion of researchers made their codelists publicly available, there is a need to encourage researchers to share their codelists. This will expedite inequalities research and prevent duplication of work. Second, while the percentages of ethnicity found in the population closely mirrored the frequencies reported in the UK census 2021,[18] we found that other marginalised groups were under-represented in the CPRD dataset (1.00%, 1.18% and 0.02% were recorded as having LD, SMI or transgender code, respectively, compared with national estimates of 2.16%, 2.00–3.00% and 0.20%[36–38]). Further research is needed to understand why these frequencies do not match national estimates.

The implementation of our codelists in inequalities research will serve to simplify the complex process of identifying these groups of interest. The codelists will improve the extraction and management of diverse health information obtained from primary healthcare records, thus providing researchers with flexible tools that can be adapted to identify marginalised populations in accordance with local definitions. This will boost the efficiency with which inequalities research is conducted using EHR.

### CONCLUSIONS

The outputs from our study provide an important contribution to future healthcare research since it is tackling the topic of addressing inequalities, which is central to the NHS Long Term Plan for this decade.[2] With the combination of this work and previously published work developing codelists for marginalised populations,[11–13] the representation of inequality groups in future research using EHR will be improved. We encourage others to use and edit our codelists as they see fit and to follow our methods to develop comprehensive codelists for the identification of other groups that may be of interest to their research topics.

**Author affiliations**
[1]School of Health Sciences, University of Surrey, Guildford, UK
[2]Nuffield Department of Primary Care Health Sciences, University of Oxford, Oxford, UK
[3]Medical School, University of Exeter, Exeter, UK
[4]Faculty of Health and Life Sciences, Northumbria University, Newcastle upon Tyne, UK
[5]Department of Behavioural Science and Health, University College London, London, UK
[6]Data Science Department, National Physical Laboratory, Teddington, UK

**Contributors** TP: conceptualisation, methodology, formal analysis, investigation, writing, review and editing. IJdVH: formal analysis, writing, reviewing and editing. BDN: formal analysis, writing, reviewing and editing. LM: formal analysis, writing, reviewing and editing. KS: conceptualisation, methodology, writing, reviewing and editing. GL: conceptualisation, methodology, writing, reviewing and editing. AL: conceptualisation, methodology, writing, reviewing and editing. KLW: conceptualisation, writing, reviewing and editing. RK: conceptualisation, methodology, writing—original draft, supervision and funding acquisition. TP is the guarantor of this study.

**Funding** The study was funded by Breast Cancer Now (reference: 2023FebIFS1615), the National Physical Laboratory (reference: NA) and the NIHR Policy Research Programme (reference: PR-PRU-NIHR206132). The views expressed are those of the author(s) and not necessarily those of the NIHR or the Department of Health and Social Care.

**Competing interests** None declared.

**Patient and public involvement** Patients and/or the public were involved in the design, conduct, reporting or dissemination plans of this research. Refer to the Methods section for further details.

**Patient consent for publication** Not applicable.

**Ethics approval** Not applicable.

**Provenance and peer review** Not commissioned; externally peer reviewed.

**Data availability statement** Data are available in a public, open access repository. The codelists developed through this work are available from Open Science Framework: https://osf.io/8skze/

**ORCID iDs**

Tetyana Perchyk http://orcid.org/0009-0002-0273-8087
Isabella Joy de Vere Hunt http://orcid.org/0000-0002-4238-0924
Brian D Nicholson http://orcid.org/0000-0003-0661-7362
Luke Mounce http://orcid.org/0000-0002-6089-0661
Georgios Lyratzopoulos http://orcid.org/0000-0002-2873-7421
Agnieszka Lemanska http://orcid.org/0000-0003-4849-2430

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
