## [Reviewer comments · BMJ Open]

ARTICLE DETAILS

Title (Provisional)

How well are marginalised groups represented in electronic records? A codelist development project and cross-sectional analysis of UK electronic health records.

Authors

Perchyk, Tetyana; de Vere Hunt, Isabella Joy; Nicholson, Brian D; Mounce, Luke; Sykes, Kate; Lyratzopoulos, Georgios; Lemanska, Agnieszka; Whitaker, Katriina L; Kerrison, Robert

VERSION 1 - REVIEW

Reviewer	1
Name	Littman, Alyson J
Affiliation	Department of Veterans Affairs Puget Sound Health Care System, Seattle,
Date	10-Feb-2025
COI	None

Overall, this was well written paper that has the potential to have a positive impact. There are a number of strengths of the paper, but there are also some limitations and points that I suggest that they consider addressing to improve its value.

Abstract

- Design: Please define codelists for readers at first mention.

- Results:

o Because multiple different and distinct groups of interest were being studied, mentioning the number of codelists across all of the populations of interest was hard for me to interpret. Consider breaking this down by population of interest.

o The findings for ethnicity were not reported consistently with the other populations of interest. Could you instead state that the proportions of individuals from ethnic minority groups were similar?

Introduction

- I was surprised that people with a learning disability were grouped with people with autism and the rationale for this grouping. Other readers may read the abstract (or see page 4, line 3), for example, when autism is not mentioned and also be surprised by this grouping, as autism is often combined with those with neurodiversity rather than learning disabilities.

Methods

- Page 6 (learning disability) – The second paragraph notes what conditions were not considered as learning disability, but it is not clear how autism fits under the umbrella of a learning disability.

- Page 6 – transgender – while I acknowledge that the language is changing quickly, and this language may have been acceptable in the past and you are unable to change the definition from the UK Government Equality Office used in lines 34-37, please reconsider the language used in lines 40-43 as many transgender people consider this language outdated. For an excellent guide for gender diversity manuscript writing, see: Veldhuis, C. B., Cascalheira, C. J., Delucio, K., Budge, S. L., Matsuno, E., Huynh, K., Puckett, J. A., Balsam, K. F., Velez, B. L., & Galupo, M. P. (2024). Sexual orientation and gender diversity research manuscript writing guide. *Psychology of Sexual Orientation and Gender Diversity*, 11(3), 365–396.
<https://doi.org/10.1037/sgd0000722>.

Public involvement

- Please provide more information about the number of members of the public who were involved in the design of the study. and which population(s) they were members of, and more specifics on how they aided the research team.

Results

- I seemed to me that the presentation of findings for ethnic minority were not consistent with how the findings were presented for other population groups. Please consider presenting if there were more or fewer (or the same) identified using the codelists as was done for other populations of interest.

- Related to the point above, in table 3 for ethnic minority groups, present the number identified from the original codelists vs. the curated codelists.

Reviewer	2
Name	Muller, Sara
Affiliation	Keele University, Research Institute for Primary Care & Health Sciences
Date	15-Apr-2025
COI	None

It is great to see systematic work in this area, as clearly defining underserved groups will encourage more high-quality research that could improve the quality of life for significant minorities within the population.

I do however have some concerns, that I think relate mainly to the reporting of the work, rather than the development of the lists themselves. Until these concerns are addressed, it is a little difficult to tell.

I have several points, that relate to the reference to a reproducible process that others can use (section 4.2). I am very supportive of this idea in principle and the inclusion of the code lists on the GitHub site – thank you. However, I am not clear which process you intend to be reproducible. Is it the whole search and Delphi exercise or using your code lists as the basis for updating a list?

I like the process (from Watson) of defining concepts and synonyms, looking for code lists and synonyms and then deciding on the code lists by consensus. However, the synonyms, particularly for SMIs, look limited. Did you include specific SMI diagnoses when conceptualising the definition and if not, why not? It is plausible that people have a code for e.g. psychosis, but no general code for SMI. The code lists in the GitHub site include codes related to specific diagnoses. How then do the terms in table 2 relate to the searches to define code lists?

The above could be improved by adding more detail to the combination and refinement of the code lists sourced from previous studies/repositories. I couldn't work out how you did this, and I think this is a key part of the method to report. Specifically, did you just combine and review the lists, the lists now being longer than previous studies because they had all missed some codes, or did you use the terms in those lists to decide on terms to include in a search of the snomed concept terms to create new lists? For me, knowing this is important.

In section 2.2, you do say "inaccurate codes were removed or corrected, and additional codes (not previously identified) added if they were associated with the diagnostic definition of one or more eligible codes." I'm not sure this check for accuracy and adding extra codes is the same as systematically looking for codes with the same terms in the descriptors. Again, this could be clearer and I would like to know more about this process.

I also really like the idea of having a secondary list of codes for use in sensitivity analyses. However, I couldn't see in the code lists provided on the GitHub site how I would identify which codes were "core" and which for the sensitivity. Is it the ones with comments next to them in the Excel versions? Most of them say to include. Please could you clarify?

I think caution is needed in saying that having more codes/defining more people to belong to a specific group than using previous definitions makes them better definitions – they could be over inclusive. For example, there are codes in the LD list for specific learning difficulties, which your quoted definition from Public Health England specially says not to include. I don't think that using the DHSC definition means these codes should be included.

Comparing the prevalence estimates to the expected prevalence/size of these groups in the population would be more convincing. I appreciate this is potentially a bit circular, depending on the quality of the data used to calculate any published estimates.

I don't think this terminology in Section 2.4 is correct "we used the CPRD database browser" the browser is a term often used to refer to the data dictionary (used to define code lists). Did you use the "define" tool and the denominator data to calculate prevalence? Either way, I think this could be more clearly explained.

Table 3 is quite messy. Would it be better as 2 tables? The first with the current top section and ethnicity – describing the population. The second with the inequality groups and the four columns relating to that (the current middle section of the table).

It would be useful to draw attention to the need for records to be comprehensive in the first place. It would also be helpful to consider under-recording of characteristics in the primary care record and the potential mis-recording (e.g. recording a learning difficulty as a learning disability). These things are unavoidable with any code list but should be acknowledged.

VERSION 1 - AUTHOR RESPONSE

REVIEWER 1 COMMENTS. Dr. Alyson J Littman, Department of Veterans Affairs Puget Sound Health Care System, Seattle.

1. Overall, this was well written paper that has the potential to have a positive impact. There are a number of strengths of the paper, but there are also some limitations and points that I suggest that they consider addressing to improve its value.

AR. Thank you for the kind words. We have incorporated your feedback throughout our manuscript (please see below).

Abstract

1. Design: Please define codelists for readers at first mention.

AR. We have added a definition of codelists to the abstract and introduction:

Page 2. "Primary care electronic health records provide a rich source of information for inequalities research. However, the reliability and validity of the research derived from these records depends on the completeness and resolution of the codelists (i.e. collection of medical terms/codes) used to identify populations of interest."

Page 4. "These data are recorded using clinical coding systems, typically containing hundreds of thousands of concepts [7], which need to be categorised via codelists (i.e. collection of medical terms/codes) to facilitate research."

2. Results: Because multiple different and distinct groups of interest were being studied, mentioning the number of codelists across all of the populations of interest was hard for me to interpret. Consider breaking this down by population of interest.

AR. Thank you for your suggestion. We agree that combining them makes it more difficult to interpret. We have now split these up for clarity:

Page 2. *“After removal of duplicated and GP review, codelists were finalized with 325 unique codes for ethnicity, 558 for LD, 499 for SMI, and 38 for transgender.”*

3. Results: The findings for ethnicity were not reported consistently with the other populations of interest. Could you instead state that the proportions of individuals from ethnic minority groups were similar?

AR. We have added the proportions for each ethnicity for consistency.

Page 2. *“The proportions identified for ethnicity were consistent with expectations for the UK population with 71% white, 6.1% Asian, 2.7% black, 1.4% mixed, and 2.2% other.”*

Introduction

1. I was surprised that people with a learning disability were grouped with people with autism and the rationale for this grouping. Other readers may read the abstract (or see page 4, line 3), for example, when autism is not mentioned and also be surprised by this grouping, as autism is often combined with those with neurodiversity rather than learning disabilities.

AR. Thank you for bringing this to our attention. We appreciate the wording is unclear (we agree that autism is a separate group and they was excluded from our definition for learning disabilities). The CORE20Plus5 framework describes the populations together as “learning disabilities or autism” and haven’t changed the way it is described in the framework itself (so as not to misrepresent it). We have added a note in the last paragraph of the introduction, clarifying that autism is not included in our definition for learning disabilities. We have also added a sentence clarifying this point in the methods:

Page 4. *“To support future health inequalities research, we set out to develop comprehensive codelists for the identification of four CORE20plus5 groups, namely: people from ethnic minority groups, people with LDs (excluding autistic people), people with SMI and people who are transgender.”*

Page 7. *“To capture the most accurate population experiencing inequalities and to avoid possible misclassifications, the more inclusive DHSC definition was used to describe the LD population. This definition also excludes people with autism, which is a separate diagnosis (a higher proportion of people with a learning disability have autism, compared with the general population, but not everyone with autism has a learning disability).”*

Methods

1. Page 6 (learning disability) – The second paragraph notes what conditions were not considered as learning disability, but it is not clear how autism fits under the umbrella of a learning disability.

AR. Please see our response to your previous comment, above.

2. Page 6 – transgender – while I acknowledge that the language is changing quickly, and this language may have been acceptable in the past and you are unable to change the definition from the UK

Government Equality Office used in lines 34-37, please reconsider the language used in lines 40-43 as many transgender people consider this language outdated. For an excellent guide for gender diversity manuscript writing, see: Veldhuis, C. B., Cascalheira, C. J., Delucio, K., Budge, S. L., Matsuno, E., Huynh, K., Puckett, J. A., Balsam, K. F., Velez, B. L., & Galupo, M. P. (2024). Sexual orientation and gender diversity research manuscript writing guide. *Psychology of Sexual Orientation and Gender Diversity*, 11(3), 365–396. <https://doi.org/10.1037/sgd0000722>.

AR. Thank you for bringing this, and helpful paper, to our attention. We have removed the terms “male-to-female” and “female-to-male” from our manuscript:

Page 7. *“Most clinical codes do not differentiate between trans men and trans women; therefore, we did not group codes into these categories according to the direction of the transition (i.e. male to female; female to male).”*

3. Public involvement. Please provide more information about the number of members of the public who were involved in the design of the study. and which population(s) they were members of, and more specifics on how they aided the research team.

AR. Thank you for the suggestion. We have received valuable support from our PPI collaborators and we have elaborated on their involvement to further acknowledge this:

Pages 7 and 8. *“Five members of the public with lived experience of autism, severe mental illness, minoritised ethnicity and/or being transgender contributed to the study. While no one with a learning disability was included, a person with autism was able to provide perspective on the differences between autism and learning disability, and their lived experience of the two being conflated with one another.*

Initial plans and findings were shared with these collaborators during quarterly meetings. During these meetings, collaborators provided feedback on the codelists, the language used to define them, and how to handle outdated or offensive terminology (such terms were retained within the codelists, to ensure completeness, but were omitted from the manuscript, so as not to perpetuate their use in the medical literature).”

Results

1. It seemed to me that the presentation of findings for ethnic minority were not consistent with how the findings were presented for other population groups. Please consider presenting if there were more or fewer (or the same) identified using the codelists as was done for other populations of interest.

AR. The purpose for inclusion of ethnicity was different than the other groups. We model that the ethnicity follows expectations from UK census estimates, and if we were to find greater proportion of different ethnicities than the expected percentages, we would be overestimating/underestimating the level of ethnic variation in the population. We have now split table 3 into two tables (see comment below) and hope that this clarifies that ethnicity was evaluated differently from the other groups.

2. Related to the point above, in table 3 for ethnic minority groups, present the number identified from the original codelists vs. the curated codelists.

AR. We have split table 3 into two tables as recommended by the second reviewer: table 3 showing the population descriptives with ethnicity, and table 4 showing increased capture with LD, SMI and transgender. We hope that separating the tables will improve presentation of the results and clarify that ethnicity was evaluated using different criteria than the other groups. (see **Page 13 of manuscript**)

REVIEWER 2. Dr. Sara Muller, Keele University

1. It is great to see systematic work in this area, as clearly defining underserved groups will encourage more high-quality research that could improve the quality of life for significant minorities within the population.

I do however have some concerns, that I think relate mainly to the reporting of the work, rather than the development of the lists themselves. Until these concerns are addressed, it is a little difficult to tell.

I have several points, that relate to the reference to a reproducible process that others can use (section 4.2). I am very supportive of this idea in principle and the inclusion of the code lists on the GitHub site – thank you. However, I am not clear which process you intend to be reproducible. Is it the whole search and Delphi exercise or using your code lists as the basis for updating a list?

AR. Thank you for your kind words and positive opinion of the importance of this work. We agree there is scope to improve the clarity of the work and have made a number of changes to achieve this.

With regards to the process we intend to be reproducible, we agree this is not clear. We followed the method described by Watson et al. As such, the approach has already been described in detail, and it was not the aim of our manuscript to describe a new methodology. The novel component of our manuscript is the curation of comprehensive codelists, which others can adopt/adapt in their own research. We've revised our manuscript to make this point clearer throughout. For example we removed the following sentence, where we believe much of the ambiguity surrounding this point originates:

Page 14. *~~"Outlining this process aims to not only create comprehensive codelists for use in inequalities research, but also helps future researchers follow these reproducible methods in generating codelists and making them available for others to use."~~*

2. I like the process (from Watson) of defining concepts and synonyms, looking for code lists and synonyms and then deciding on the code lists by consensus. However, the synonyms, particularly for SMIs, look limited. Did you include specific SMI diagnoses when conceptualising the definition and if not, why not? It is plausible that people have a code for e.g. psychosis, but no general code for SMI. The code lists in the GitHub site include codes related to specific diagnoses. How then do the terms in table 2 relate to the searches to define code lists?

AR. Thank you for bringing this important omission to our attention. This was an oversight on our part. We have since gone back to the CPRD bibliography and searched for additional papers using the terms: 'Schiz*', 'Psychosis,' 'Bipolar' and 'Major depress*.' We identified an additional 15 papers using these terms (nine, three, two and one, respectively). Of these papers, 3 included code lists for the specific conditions. In total, 9 codes were identified as not present in our original list and were reviewed using our SMI definition and subsequently sent for GP review. It was determined that all 9 codes were to be excluded from our final codelist. This is because they were not conclusive of SMI (e.g. drug induced

episodes). We have updated the results accordingly to indicate the additional papers and codes reviewed, added the additional search terms to table 2, updated the numbers in figure 1, and updated the numbers in the text of the results. (See **page 8 and 10 of manuscript**)

3. The above could be improved by adding more detail to the combination and refinement of the code lists sourced from previous studies/repositories. I couldn't work out how you did this, and I think this is a key part of the method to report. Specifically, did you just combine and review the lists, the lists now being longer than previous studies because they had all missed some codes, or did you use the terms in those lists to decide on terms to include in a search of the snomed concept terms to create new lists? For me, knowing this is important.

AR. Thank you for your feedback. We have elaborated on the process to clarify the steps taken, including how the codelists were combined and how they were reviewed with the Code Browser.

Page 9. *“Once all relevant published codelists were identified, they were combined into one excel file for each inequality group. This gave us four files: one for ethnicity, LD, SMI, and transgender. Within each of these files, any duplicate codes were removed, and remaining codes grouped by the two main coding systems used in the UK: Read and systematized nomenclature of medicine clinical terminology (SNOMED-CT). Read was generally used in primary care until 2018, and SNOMED-CT thereafter, following the enablement of SNOMED-CT within Primary care systems [30]. This provided an aggregated list encompassing every code (excluding duplicates) that has been used and published wither through CPRD literature or published through an online codelist database.*”

These aggregated codelists were then checked for accuracy using the CPRD Code Browser, which is a tool specific to CPRD and is provided to researchers who gain access to their data. This tool contains the diagnostic description, the alphanumeric Read / SNOMED-CT code originally used by the general practitioner (GP) to enter the clinical information, and CPRD's proprietary 'medcode' / 'medcodeid' (which is simply a numeric equivalent of the Read / SNOMED-CT code) [30]. The tool allowed for easy browsing by either using a known code or description to check code accuracy. Searching by the medical descriptors for the code, or “terms,” was useful because each search provided a list of all possible codes. For example, a search for “psychosis” showed every diagnostic term and associated code available from primary care GP records. This was useful to cross reference each of the individual codes and ensure all relevant options have been included. In line with the method described by Watson et al [14], inaccurate codes were removed or corrected, and additional codes (not previously identified) added if they were associated with the diagnostic definition of one or more eligible codes.”

4. In section 2.2, you do say “inaccurate codes were removed or corrected, and additional codes (not previously identified) added if they were associated with the diagnostic definition of one or more eligible codes.” I'm not sure this check for accuracy and adding extra codes is the same as systematically looking for codes with the same terms in the descriptors. Again, this could be clearer and I would like to know more about this process.

AR. Thank you for your feedback. We have addressed this comment, alongside your previous comment. Please see above edits for clarification.

4. I also really like the idea of having a secondary list of codes for use in sensitivity analyses. However, I couldn't see in the code lists provided on the GitHub site how I would identify which codes were “core” and which for the sensitivity. Is it the ones with comments next to them in the Excel versions? Most of them say to include. Please could you clarify?

AR. Thank you for bringing this to our attention. We have moved all the sensitivity codes to the bottom of each list and indicated the column as “sensitivity” to show that these are for sensitivity analyses.

Please see updates on <https://osf.io/8skze/>

5. I think caution is needed in saying that having more codes/defining more people to belong to a specific group than using previous definitions makes them better definitions – they could be over inclusive. For example, there are codes in the LD list for specific learning difficulties, which your quoted definition from Public Health England specially says not to include. I don’t think that using the DHSC definition means these codes should be included. Comparing the prevalence estimates to the expected prevalence/size of these groups in the population would be more convincing. I appreciate this is potentially a bit circular, depending on the quality of the data used to calculate any published estimates.

AR. We agree it is erroneous to conclude that, because our codelists identify more people who could be described as having learning disabilities (depending on the definition used), compared with other codelists, they are therefore ‘better.’ We have, therefore, revised our conclusions to state:

Page 15. *“The codelists will improve the extraction and management of diverse health information obtained from primary health care records, providing researchers with flexible tools that can be adapted to identify marginalised populations in accordance with local definitions.”*

We agree it would be more compelling to compare the number of individuals with individual conditions to national estimates (e.g. number of people with Down’s Syndrome identified using our codelist, existing codelists, and national estimates). Unfortunately, due to the sheer number of conditions, and the lack of data regarding national estimates for each of these, this is not possible. We hope that, by stating combined totals, and comparing them with national totals, we can at least highlight discrepancies between the two, potential issues with defining and identifying groups, and the need for promoting data capture.

6. I don’t think this terminology in Section 2.4 is correct “we used the CPRD database browser” the browser is a term often used to refer to the data dictionary (used to define code lists). Did you use the “define” tool and the denominator data to calculate prevalence? Either way, I think this could be more clearly explained.

AR. Thank you for bringing this typo to our attention. It should read: “we used the CPRD database”, rather than “The CPRD database browser”. We have corrected this in our manuscript:

Page 10. *“To explore the comprehensiveness of the developed codelists, we used the CPRD database to ascertain the number of people with at least one recording of an included code for each list.”*

7. Table 3 is quite messy. Would it be better as 2 tables? The first with the current top section and ethnicity – describing the population. The second with the inequality groups and the four columns relating to that (the current middle section of the table).

AR. This is a great idea, thank you. We have followed your advice for editing it into two tables and agree that it looks much cleaner split in this way. Please see **Page 13 of the manuscript** for the table edits. There were also minor edits made to the ordering of the results section to reflect this change.

8. It would be useful to draw attention to the need for records to be comprehensive in the first place. It would also be helpful to consider under-recording of characteristics in the primary care record and the potential mis-recording (e.g. recording a learning difficulty as a learning disability). These things are unavoidable with any code list but should be acknowledged.

AR. Thank you for the suggestion. We agree these are important factors to consider. We have expanded on these points in the limitations section of the discussion.

Page 14. *“Furthermore, improving on the accuracy of the codelists will not fully address some of the possible underlying issues with primary care data. For example, the decreased capture of certain inequalities compared to population estimates may reflect under recording within primary care records. For example, transgender people are likely to seek medical care through private practice and will be generally underrepresented in the primary care population. We aim for increasing research in this area to ultimately improve the data quality and understanding the impacts of inequality.”*

VERSION 2 - REVIEW

Reviewer	1
Name	Littman, Alyson J
Affiliation	Department of Veterans Affairs Puget Sound Health Care System, Seattle,
Date	20-Jun-2025
COI	

The authors were responsive to my prior comments and suggestions and the manuscript is much improved. I found some of the revisions regarding whether autism was or was not included as part of the learning disability grouping confusing. It would be helpful to explicitly state that no autism diagnoses were included in the LD codelists, and avoid parenthetical statements.

Introduction

- Page 4, Paragraph 2, line 1 – consider adding the word “eliminating” or “addressing” so the sentence reads “in the UK, eliminating health inequalities is a national priority”

- Page 4, paragraph 4, line 3 – I was confused by the parenthetical statement “excluding autistic people.” Instead of making this a parenthetical statement in the introduction, I would spell this out clearly in the methods.

Methods

- Page 7, paragraph 2

o I would remove the word “also” in the statement “This definition also excludes autism” because there were no exclusions discussed previously in this paragraph.

o I suggest deleting the parenthetical statement about the overlap between learning disability and autism. I found it more confusing to add that.

Other minor suggestions:

- Page 9, paragraph 2 – I think “filed” should be files and “wither” should be whether.

- Page 9 under Step 3. Since the first step includes only one author, unless that author’s pronouns are they/them, it seems like it should be “she” (as in she scored each code).

Reviewer	2
Name	Muller, Sara
Affiliation	Keele University, Research Institute for Primary Care & Health Sciences
Date	23-Jun-2025
COI	

This is much improved and I really welcome this work.

I have a minor suggestion for the abstract.

You rightly explains that accurate code lists are required for high quality research using routinely collected data such as CPRD. I know words will be tight, but it would be helpful to mention that the data also need to be recorded. The low prevalence of LD, for example, exemplifies that this isn't always the case and others should be aware that a good code list can't find diagnoses that were not entered.

VERSION 2 - AUTHOR RESPONSE

REVIEWER 1 COMMENTS. Dr. Alyson J Littman, Department of Veterans Affairs Puget Sound Health Care System, Seattle.

Comments to the Author:

The authors were responsive to my prior comments and suggestions and the manuscript is much improved. I found some of the revisions regarding whether autism was or was not included as part of the learning disability grouping confusing. It would be helpful to explicitly state that no autism diagnoses were included in the LD codelists, and avoid parenthetical statements.

AR. Thank you for the positive comments. We found your feedback constructive and agree the paper is much improved for the changes made.

With regards to explicitly stating that no autism diagnoses were included in the LD codelists, and avoiding parenthetical statements, we have rewritten the relevant section in the Methods as follows:

Page 8. "To capture the most accurate population experiencing inequalities and to avoid possible misclassifications, the more inclusive DHSC definition was used to describe the LD population. This definition for learning disability excludes autism, which is important, as while many people with a LD have autism, not everyone with autism has a learning disability. Accordingly, no autism codes were included in the LD codelist."

Introduction

1. Page 4, Paragraph 2, line 1 – consider adding the word “eliminating” or “addressing” so the sentence reads “in the UK, eliminating health inequalities is a national priority”

AR. Thank you for the suggestion. We can see how this small addition improves the sentence, and have added the term “addressing” accordingly:

Page 5. "In the United Kingdom (UK), addressing health inequalities are a national priority [2]."

2. Page 4, paragraph 4, line 3 – I was confused by the parenthetical statement “excluding autistic people.” Instead of making this a parenthetical statement in the introduction, I would spell this out clearly in the methods.

AR. Thank you for bringing this to our attention. We can see how this might lead to confusion and agree it would be better to explain the point in the methods section, specifically. We have deleted the parenthetical statement regarding autism, in addition to spelling out what we did more clearly in the methods (please see previous change above). The introduction now reads as follows:

Page 5. "To support future health inequalities research, we set out to develop comprehensive codelists for the identification of four CORE20plus5 groups, namely: people from ethnic minority groups, people with LDs, people with SMI and people who are transgender."

Results

3. Page 7, paragraph 2:

o I would remove the word “also” in the statement “This definition also excludes autism” because there were no exclusions discussed previously in this paragraph.

o I suggest deleting the parenthetical statement about the overlap between learning disability and autism. I found it more confusing to add that.

AR. Thank you again for raising this important point. Please see our previous responses for details regarding how we have revised our manuscript in accordance with your comments.

Other minor suggestions

4. Page 9, paragraph 2 – I think “filed” should be files and “wither” should be whether.

AR. Thank you for pointing out these spelling errors. We have corrected accordingly.

5. Page 9 under Step 3. Since the first step includes only one author, unless that author’s pronouns are they/them, it seems like it should be “she” (as in she scored each code).

AR. Thank you for raising this potentially confusing point. We have changed the wording to avoid using pronouns:

Page 10. *“The codes were scored using the following three-point scale:”*

REVIEWER 2. Dr. Sara Muller, Keele University

Comments to the Author:

This is much improved, and I really welcome this work.

AR. Thank you for the kind words. We are glad our revisions have been well received and would like to thank you for your previous review, which really helped us to refine the paper.

1. I have a minor suggestion for the abstract. You rightly explain that accurate code lists are required for high quality research using routinely collected data such as CPRD. I know words will be tight, but it would be helpful to mention that the data also need to be recorded. The low prevalence of LD, for example, exemplifies that this isn't always the case and others should be aware that a good code list can't find diagnoses that were not entered.

AR. Thank you for the suggestion. We agree this is an important point to mention. We added a statement to the ‘Conclusion’ section of the Abstract:

Page 3. *“The curated codelists are more sensitive than those widely used in practice and research. Discrepancies between national estimates and primary care records suggest potential recording/retention issues. Resolving these requires further investigation and could lead to improved data quality for research.”*